# *Bmapaf-1* is Involved in the Response against BmNPV Infection by the Mitochondrial Apoptosis Pathway

**DOI:** 10.3390/insects11090647

**Published:** 2020-09-22

**Authors:** Xue-yang Wang, Xin-yi Ding, Qian-ying Chen, Kai-xiang Zhang, Chun-xiao Zhao, Xu-dong Tang, Yang-chun Wu, Mu-wang Li

**Affiliations:** 1Jiangsu Key Laboratory of Sericultural Biology and Biotechnology, School of Biotechnology, Jiangsu University of Science and Technology, Zhenjiang 212018, China; xueyangwang@just.edu.cn (X.-y.W.); ding_xin_yi@163.com (X.-y.D.); 15751771696@163.com (Q.-y.C.); zhangjustedu@163.com (K.-x.Z.); www_zcx123456@163.com (C.-x.Z.); xudongt@just.edu.cn (X.-d.T.); 2Key Laboratory of Silkworm and Mulberry Genetic Improvement, Ministry of Agriculture and Rural Affairs, Sericultural Research Institute, Chinese Academy of Agricultural Science, Zhenjiang 212018, China

**Keywords:** *Bombyx mori*, BmNPV, *apoptotic protease-activating factor 1*, mitochondrial apoptosis pathway, response analysis

## Abstract

**Simple Summary:**

Apaf-1 is involved in the apoptosis pathway and *Bmapaf-1* showed a significant response to BmNPV infection in our previous transcriptome data. In this study, the underlying mechanism of *Bmapaf-1* in response to BmNPV infection was studied. To preliminarily determine the relationship of *Bmapaf-1* with BmNPV, the expression pattern of *Bmapaf-1* was analyzed in different tissues of differentially resistant silkworm strains following virus infection. To further define the role of *Bmapaf-1* in BmNPV infection, the alteration of BmNPV infection in BmN cells and the expression patterns of *Bmcas-Nc* and *Bmcas-1* were analyzed following knockdown and overexpression of *Bmapaf-1* using siRNA and the pIZT/V5-His-mCherry insect vector, respectively. Furthermore, to analyze whether *Bmapaf-1* is involved in BmNPV infection by apoptosis, the inducer NSC348884 and inhibitor Z-DEVD-FMK were used.

**Abstract:**

Discovery of the anti-BmNPV (*Bombyx mori* nuclearpolyhedrovirus) silkworm strain suggests that some kind of antiviral molecular mechanism does exist but is still unclear. Apoptosis, as an innate part of the immune system, plays an important role in the response against pathogen infections and may be involved in the anti-BmNPV infection. Several candidate genes involved in the mitochondrial apoptosis pathway were identified from our previous study. *Bombyx mori apoptosis protease-activating factor-1* (*Bmapaf-1*) was one of them, but the antiviral mechanism is still unclear. In this study, sequences of BmApaf-1 were characterized. It was found to contain a unique transposase_1 functional domain and share high CARD and NB-ARC domains with other species. Relatively high expression levels of *Bmapaf-1* were found at key moments of embryonic development, metamorphosis, and reproductive development. Further, the significant difference in expression of *Bmapaf-1* in different tissues following virus infection indicated its close relationship with BmNPV, which was further validated by RNAi and overexpression in BmN cells. Briefly, infection of budded virus with enhanced green fluorescent protein (BV-EGFP) was significantly inhibited at 72 h after overexpression of *Bmapaf-1*, which was confirmed after knockdown of *Bmapaf-1* with siRNA. Moreover, the downstream genes of *Bmapaf-1*, including *Bmnedd2-like caspase* (*BmNc*) and *Bmcaspase-1* (*Bmcas-1*), were upregulated after overexpression of *Bmapaf-1* in BmN cells, which was consistent with the RNAi results. Furthermore, the phenomenon of *Bmapaf-1* in response to BmNPV infection was determined to be related to apoptosis using the apoptosis inducer NSC348884 and inhibitor Z-DEVD-FMK. Therefore, *Bmapaf-1* is involved in the response against BmNPV infection by the mitochondrial apoptosis pathway. This result provides valuable data for clarifying the anti-BmNPV mechanism of silkworms and breeding of resistant silkworm strains.

## 1. Introduction

Sericulture has existed in China for more than 5000 years and is the main income for farmers who rear silkworms. BmNPV is a double-stranded DNA virus that causes serious economic losses ever year. Many silkworm strains have been found to have a high resistance to BmNPV infection [1,2], but the molecular mechanism of silkworm resistance to BmNPV is still unclear. With the fast development of biotechnology, many new technologies have been used to study the antiviral mechanism of silkworms, such as the RNA-seq transcriptome [3,4], isobaric tag for relative and absolute quantification (iTRAQ), and label-free proteomics [5,6]. Plenty of candidate genes and proteins involved in the response to BmNPV have been identified, but the functions of most of them are still unknown and require further validation in viral infection.

Apoptosis, also called programmed cell death (PCD), is a physiological process in pluricellular organisms. A significant characteristic of apoptosis is the removal of unwanted and potentially dangerous cells [7], which has been widely reported to play an important role in defense against viral infection [8]. A relatively substantial amount of evidence shows that mitochondria are one of the major organelles involved in signal transduction and activation of cell death [9]. The death of stimulated cells by apoptosis is triggered by proteins released from the mitochondrial intermembrane space, such as cytochrome c (cytc), and this released protein interacts with Apaf-1 and caspase-9 to form the apoptosome [10,11,12]. This pathway is known as the mitochondrial apoptosis pathway, which plays a central role in regulating mammalian cell apoptosis [13]. Once cytc is released into the cytosol, it binds with Apaf-1, and this permits the binding of deoxyadenosine triphosphate (dATP) or adenosine triphosphate (ATP), triggering its oligomerization to form the apoptosome [14,15]. After recruitment of multiple procaspase-9 molecules, autoactivation will start. Additionally, executioner caspase-3 can be activated by cleaved caspase-9, and apoptosis can proceed [16,17].

In our previous RNA-seq transcriptome analysis, among the midgut of different silkworm resistant strains following BmNPV infection, several candidate genes belonging to the mitochondrial apoptotic pathway were identified to be significantly differentially expressed after virus infection, including *Bmapaf-1*, *Bmcytc*, *Bmcas-1*, and *BmNc* [3]. It is little wonder that *Drosophila* contains the canonical apoptosome protein Apaf-1 (also known as CED4) and appears to use it. Apaf-1 is similar to that which has been described for mammalian cells and can activate downstream effector caspases by combining with caspase-9 and cytc [18,19]. It also has a close relationship with viral infection; the activation of Apaf-1 and caspase-9 in immortalized human hepatocytes was reported to inhibit core protein expression of the hepatitis C virus [20]. However, the role of BmApaf-1 in silkworms is still unknown. In this study, we explored the role of Bmapaf-1 in response to BmNPV infection in order to clarify the mechanism of the silkworm response to BmNPV.

To further study the antiviral mechanism of *Bmapaf-1* in response to BmNPV, the expression of *Bmapaf-1* was knocked down and overexpressed using siRNA and the pIZT-His-mCherry vector, respectively. The variation of BmNPV was recorded and determined using fluorescence microscopy and RT-qPCR, respectively. Moreover, the relationship of *Bmapaf-1* and apoptosis was analyzed using the apoptosis inducer, NSC348884, and inhibitor, Z-DEVD-FMK.

## 2. Materials and Methods

### 2.1. Silkworm and BmNPV

The susceptible strain, YeB, and the resistant strain, YeA, and p50 (no relationship with YeA and YeB) were maintained in the Key Laboratory of Sericulture, School of Life Sciences, Jiangsu University of Science and Technology University, Zhenjiang, China. The first three instars silkworm larvae were fed with fresh mulberry leaves at 26 ± 1 °C, 75 ± 5% relative humidity, and a 12-h day/night cycle. The rearing temperature was reduced to 24 ± 1 °C for the last two instars, and the other conditions were the same with the first three instar larvae.

Budded virus of BmNPV containing enhanced green fluorescent protein tag (BV-EGFP) was generously donated by Professor Xu-dong Tang and was kept in our laboratory. The EGFP gene was inserted into the plasmid pFASTbac1 using the *Bam*H I-*Xho* I site to generate a recombinant viral vector to express the EGFP protein under the drive of the polyhedron promoter. The titer of BV-EGFP (pfu/mL) was calculated using the method as described in our previous study [21]. The culture containing BV-EGFP (1 × 10^8^ pfu/mL) was used to infect BmN cells between different groups, and the control group was treated with equal volume culture.

### 2.2. Bioinformatics Analysis

The coding sequence (CDS) and the deduced amino acid sequences of BmApaf-1 (ID: NP_001186937.1) were aligned using the DNAMAN 8.0 software (LynnonBiosoft, San Ramon, CA, USA). The conserved motif of BmApaf-1 amino acid was predicted using the SMART server (http://smart.embl-heidelberg.de/). The analysis of homologous sequences of Apaf-1 in different species was conducted using the BLASTP tool (http://www.ncbi.nlm.nih.gov/). Amino acid sequence of BmApaf-1 and its homologs among different species were aligned using the MEGA-X software with the MUSCLE module. A neighbor-joining tree was generated using MEGA-X software with a bootstrap of 1000 replications and the LG+G DNA/Protein model.

### 2.3. Sample Preparation, RNA Extraction, and cDNA Synthesis

Each of the first day of fifth instar larva of YeA and YeB was inoculated with 2 μL of culture containing BV-EGFP (1.0 × 10^8^ pfu/mL) and then reared under standard conditions. The samples were prepared after BV-EGFP infection at 48 h. The midgut, hemolymph, fat body, and malpighian tubule of larva were dissected and washed with DEPC water, and 30 samples of each tissue were mixed together to minimize individual genetic differences. The whole bodies of 30 larvae at different developmental stages were mixed. Each experiment was repeated using three biological replicates. All samples were quick-frozen using liquid nitrogen, and then stored at −80 °C until use.

The total RNA of different silkworm tissues and BmN cells was isolated using TRIzol Reagent (Invitrogen, California, USA) following the manufacturer’s instructions. The concentrations and purities of RNA were quantified using the NanoDrop 2000 spectrophotometer (Thermo Fisher Scientific, Waltham, MA, USA). The RNA integrity was verified using a 1% agarose gel for electrophoresis. The first strand cDNA was synthesized with 1.0 μg RNA using the PrimeScript ^TM^ RT reagent Kit with gDNA Eraser (Perfect Real Time; TaKaRa, Kusatsu, Japan) following the manufacturer’s instructions. The quality of the synthetic cDNA was verified using the reference gene, *B. mori glyceraldehyde-3-phosphate dehydrogenase* (*BmGAPDH*).

### 2.4. RT-qPCR

The expression level of genes in this study was analyzed using RT-qPCR. All primer sequences with high amplification efficiency (≥95%) are listed in Table 1. RT-qPCR reactions were prepared with the NovoStart ^®^SYBR qPCR SuperMix Plus (Novoprotein, Shanghai, China) following the manufacturer’s instructions. Briefly, each 15 μL of reaction volume contained 2.0 μL of cDNA, 7.5 μL of SYBR SuperMix, 0.6 μL of each gene-specific primer (0.4 μM), and 4.3 μL of ddH_2_O. Reactions were carried out using the LightCycler^®^ 96 System (Roche, Basel, Switzerland). The thermal cycling program included an initial denaturation at 95 °C for 5 min, and then 40 cycles under 95 °C for 5 s and 60 °C for 31 s. All measurements were repeated in triplicate. The 2^−△△Ct^ method was adopted to calculate the relative expression level, based on the method described by Livak et al. (2001). *BmGAPDH* was used as an internal control [22]. Differences among the three repeats were analyzed using the SPSS Statistics 20 software (IBM, Endicott, NY, USA) with the one-way ANOVA method. A *p*-value less than 0.05 was considered as statistically significant among different groups.

### 2.5. Synthesis of siRNA

To knockdown the expression of *Bmapaf-1* in BmN cells, two specific siRNAs targeting sequences located in the functional domain of *Bmapaf-1* were selected and designed using the method in a previous study [23]. The target DNA sequence was inserted behind the T7 promoter, and the siRNA oligos were synthesized by SUNYA Biotechnology (Zhejiang, China; Table 2). The siRNA oligos were used to synthesize the template, which could transcript into siRNA using an In Vitro Transcription T7 Kit (for siRNA synthesis; TaKaRa, Japan) according to the manufacturer’s instructions. The concentration and purity of siRNAs were detected by the NanoDrop 2000 spectrophotometer (Thermo Fisher Scientific, Waltham, MA, USA). The integrity of siRNAs was determined by 3% agarose gel electrophoresis. The newly synthesized siRNA with good quality was stored at −80 °C until use.

### 2.6. Construction of pIZT-mCherry-Bmapaf-1 Overexpression Vector

The functional domain of *Bmapaf-1* was amplified from the cDNA of BmN cells with the *Bmapaf-1KE* primers (Table 1; the underline indicates *Kpn* I or *Eco*R I restriction sites). The purified PCR products were cloned into a pMD-19T vector for sequencing. The pMD-19T-Bmapaf-1 and the pIZT/V5-His-mCherry vector were digested with *Kpn* I and *Eco*R I (TaKaRa, Japan), and then ligated with T_4_ DNA ligase (TaKaRa, Kusatsu, Japan) at 16 °C overnight. Positive colonies were verified using PCR, and the recombinant expression vector pIZT/V5-His-mCherry-Bmapaf-1 was further verified by *Kpn* I and *Eco*R I digestion and sequenced by SUNYA Biotechnology (Zhejiang, China).

### 2.7. BmN Cell Culture, Transfection, and Fluorescence Signal Acquisition

The BmN cells derived from the silkworm ovary were cultured in TC-100 (AppliChem, Gatersleben, Germany) that contained 10% (*v/v*) fetal bovine serum (FBS; Thermo Fisher Scientific, Waltham, MA, USA) and 1% double-antibiotics (penicillin and streptomycin) at 28 °C [24]. The siRNA and overexpression vector of *Bmapaf-1* were transfected into BmN cells using the Neofect ^TM^ DNA transfection reagent (NEOFECT, Beijing, China). Briefly, BmN cells were seeded into 30 mm culture flasks (approximately 1 × 10^5^ cells/well) before transfection. Then, 4.0 µg of siRNA or vector and 4.0 µL of transfection reagent were added successively into 200 µL of serum-free TC-100 to prepare the transfection solution, which was added into the culture medium to finish the transfection. The best efficiency was obtained at 24 h, and this time point was selected for all subsequent analysis.

The fluorescence signal in this study was captured at 24, 48, 72 h post-inoculation after knockdown or overexpression of *Bmapaf-1* at 24 h using an inverted microscope DMi3000B camera (Leica, Solms, Germany), and the picture was processed with the Application Suite V4.6 software (Leica, Germany).

### 2.8. Inhibition and Induction of Apoptosis

Z-DEVD-FMK and NSC348884 reagents (Topscience; Beijing, China) were used for inhibition and induction of apoptosis, respectively. Both of these compounds were dissolved in DMSO to generate a 1 mM working solution. The final concentrations of 5 µM and 10 µM were selected for NSC348884 and Z-DEVD-FMK, respectively, based on the gradient concentration test on BmN cells. Inhibition and induction effects in this study were analyzed at 72 h after treatment with Z-DEVD-FMK and NSC348884.

## 3. Results

### 3.1. Characterization of the BmApaf-1 Sequence

The open reading fragment (ORF) of *Bmapaf-1* (GenBank ID: NM_001200008.1) contained a complete 4302 bp, which encoded a 1433-amino acid protein. The theoretical isoelectric point (*p*I) and molecular weight (MW) were 7.04 and 16.30 kDa, respectively. BmApaf-1 protein contained three major functional domains: transposase_1, caspase recruitment domain (CARD), and a signaling motif NB-ARC (Appendix A).

BLASTP blast results showed that the amino acid sequence of BmApaf-1 was most similar to that of *Bombyx mandarina* (XP_028036457.1, 92.48% identity), followed by *Manduca sexta* (XP_030022507.1, 58.33%), *Heliothis virescens* (PCG76501.1, 52.45%), *Trichoplusia ni* (XP_026734098.1, 55.59%), *Helicoverpa armigera* (XP_021181657.1, 52.02%), and *Papilio machaon* (XP_014359375.1, 52.15%). A homologous alignment showed that a unique functional domain, transposase_1, was found in the BmApaf-1 amino acid sequence, compared to its homologs in other species, indicating BmApaf-1 might have an unknown and special role (Appendix A). Moreover, BmApaf-1 amino acid sequence shared high similarity in CARD and NB-ARC domains among different species; this indicated that BmApaf-1 might play an important role in the silkworm apoptosis pathway.

To determine the phylogenetic tree of Apaf-1 among different species, CDS sequences of BmApaf-1 and its homologs from other species were downloaded from NCBI. Accession numbers of BmApaf-1 homologs are listed in Appendix A. A phylogenetic tree that contained BmApaf-1 and 15 other homologs was generated based on the DNA/protein model of JTT+G using MEGA X software (Appendix A). BmApaf-1 shared a low sequence similarity with its counterparts from *Pieris rapae*, *Vanessa tameamea*, and *Bicyclus anynana*, indicating that Apaf-1 might become divergent among Lepidoptera.

### 3.2. The Spatiotemporal Expression Pattern of Bmapaf-1

The silkworm p50 is a strain that is widely used in different laboratories, and its genome is available online. To preliminarily determine the biological function of *Bmapaf-1*, the relative expression level of *Bmapaf-1* at different stages and different tissues of the p50 strain were detected by RT-qPCR. Among different development times of the egg, the highest expression level of *Bmapaf-1* was found at the period when protuberance occurred (Figure 1A, 2nd day). Among the different development stages, the highest level was detected at the adult stage (Figure 1B), and the relatively high expression levels were observed in the testis and ovary (Figure 1C).

### 3.3. Bmapaf-1 Showed Significant Response to BmNPV Infection in Different Tissues

To preliminarily determine the relationship of *Bmapaf-1* with BmNPV infection, the expression pattern of *Bmapaf-1* was tested in different tissues of YeA (resistant strain) and YeB (susceptible strain) following BmNPV inoculation at 48 h, including the midgut, hemolymph, fat body, and malpighian tubule. The negative control is injected with BmN cell culture medium, and the blank control is given no injection. The resistant levels of YeA and YeB were tested in our previous report [2]. The medial lethal concentration (LC_50_) value of YeA was more than 10^9^ OB/mL, but p50 was just about 10^5^ OB/mL. Results showed that *Bmapaf-1* had a reverse expression trend in the two different resistant strains, except in the midgut (Figure 2). *Bmapaf-1* had a significantly higher expression level in selected tissues of YeA following virus infection, except in the hemolymph. Moreover, it had upregulated expression levels in the midgut and hemolymph of YeB following BmNPV infection and downregulated expression levels in the others (Figure 2). Generally, the significantly different expression levels of *Bmapaf-1* in the two strains, following virus infection, indicated its vital role in response to BmNPV.

### 3.4. Selected Downstream Genes Were Downregulated after Knockdown of Bmapaf-1 in BmN Cells

To further study the role of *Bmapaf-1* in response to BmNPV infection, two siRNAs targeting the functional domain of *Bmapaf-1* were used to knockdown the expression of *Bmapaf-1* in BmN cells. The preliminary experiment demonstrated that 4 μg of the siRNA targeting *Bmapaf-1* (siapaf-1) was effective for interfering with the expression of *Bmapaf-1* in BmN cells (data not shown). Expression levels of *Bmapaf-1* were analyzed after transfecting with siapaf-1 at different times using RT-qPCR. The siRNA targeting red fluorescence protein (siRFP) was used as a negative control. Results showed that *Bmapaf-1* was significantly downregulated after transfection with siapaf-1at 72 h, and this time point was selected for further analysis (Figure 3A).

In this study, two downstream genes of *Bmapaf-1*, i.e., *BmNc* and *Bmcas-1*, were selected by analyzing the mitochondrial apoptosis pathway of *Drosophila*. To get the relationship of *Bmapaf-1* with *BmNc* and *Bmcas-1*, expression levels of the two genes were analyzed after transfection with siapaf-1 at different time points in BmN cells. *BmNc* showed a significant upregulation at 24 h post siapaf-1 transfection, and quickly downregulated at 48 and 72 h (Figure 3B). *Bmcas-1* showed a significant downregulation at 48 h post siapaf-1 transfection (Figure 3C). These results indicated that *Bmapaf-1* had a close relationship with the expression of *BmNc* and *Bmcas-1*.

### 3.5. Knockdown of Bmapaf-1 Promoted BmNPV Infection in BmN Cells

To confirm the role of *Bmapaf-1* during BmNPV infection, 20 μL of the culture medium containing BV-EGFP (1 × 10^8^ pfu/mL) was added into the BmN cells (30 mm dish) that had been transfected with siapaf-1 for 24 h. BmNPV infection signal was collected using fluorescence microscopy at 24, 48, and 72 h after inoculation with BV-EGFP and detected using RT-qPCR. The group transfected with siRFP was used as a negative control. The green fluorescence signal of the virus was significantly higher in the group treated with siapaf-1 at 72 h compared with the control (Figure 4C), but no difference was observed at 24 and 48 h (Figure 4A,B). To further validate the phenomenon, the capsid gene *vp39* of BmNPV was used to determine viral replication in different groups using RT-qPCR. The expression level of *vp39* was significantly higher in the siapaf-1 group than that in the control at 72 h (Figure 4D), but no significant difference was found at 24 and 48 h, which was consistent with the fluorescent images described above. These data indicated that *Bmapaf-1* played an important role in response against BmNPV infection.

### 3.6. Overexpression of Bmapaf-1 Upregulated the Expression of Its Downstream Genes in BmN Cells

To overexpress *Bmapaf-1* in BmN cells, a recombinant plasmid, pIZT-mCherry-Bmapaf-1, was constructed using double enzyme cutting (Figure 5B). The functional domain of *Bmapaf-1* was inserted into the pIZT-mCherry vector between *Kpn* I and *EcoR* I. Subsequently, the recombinant plasmid was transfected into BmN cells using the Neofect ^TM^ DNA transfection reagent according to manufacturer’s instructions. The red fluorescence signal was observed using fluorescence microscope, and results indicated that pIZT-mCherry-Bmapaf-1 was transfected into BmN cells and successfully expressed (Figure 5A). It was also validated by the 30-fold upregulation of *Bmapaf-1* in the transgenic BmN cell line using RT-qPCR, compared to the control group (Figure 5C).

To further validate the relationship of *Bmapaf-1* with *BmNc* and *Bmcas-1*, expression levels of the two genes were determined between the transgenic BmN cell line and negative group using RT-qPCR. The expression level of *BmNc* in the transgenic cell line was more than two times higher than that in the control group (Figure 5D), and *Bmcas-1* even reached a seven-fold enhancement in the transgenic cell line as compared with the negative group (Figure 5E). These results further validated the regulation relationship of *Bmapaf-1* to *BmNc* and *Bmcas-1*.

### 3.7. Overexpression of Bmapaf-1 Inhibited BmNPV Infection in BmN Cells

To further validate the role of *Bmapaf-1* in response to BmNPV infection after RNAi, the variation of BmNPV infection was recorded in the transgenic cell line after inoculation with BV-EGFP using fluorescence microscopy at different time points. pIZT-mCherry was used as negative control. The green fluorescence signal of the virus was significantly lower in the transgenic cell line at 72 h post-inoculation compared with the negative control (Figure 6C), but no difference was observed at 24 and 48 h (Figure 6A,B). To further validate the result of Figure 6A–C, the capsid gene *vp39* of BmNPV was analyzed to evaluate the viral replication in different groups using RT-qPCR. The expression level of *vp39* was significantly downregulated in the transgenic cell line at 72 h compared to the control group (Figure 4D), but no significant difference was observed at 24 and 48 h, which was consistent with the fluorescent images described above. These data indicated that *Bmapaf-1* played a vital role in anti-BmNPV infection.

### 3.8. Apoptosis Regulated by Bmapaf-1 Involved in Response against BmNPV ISnfection

To analyze whether *Bmapaf-1* was involved in BmNPV infection by regulating apoptosis, the variation of BmNPV replication was analyzed in the RNAi and overexpression group after treatment with the apoptosis inducer, NSC348884, and inhibitor, Z-DEVD-FMK, using RT-qPCR, respectively. BmNPV infection significantly increased after knockdown of *Bmapaf-1* in BmN cells, as shown in Figure 4, but the infection of BV-EGFP was significantly inhibited in the groups treated with the inducer NSC348884 at 72 h after transfection with siapaf-1and siRFP (Figure 7A). Moreover, BmNPV infection was inhibited after overexpression of *Bmapaf-1* in BmN cells, as shown in Figure 6, but the infection of BV-EGFP was significantly increased at 72 h after treatment with inhibitor Z-DEVD-FMK in the transgenic cell line and its negative control (Figure 7B). These results indicated that *Bmapaf-1* was involved in the response against BmNPV infection by the apoptosis.

## 4. Discussions

BmNPV, one of the main silkworm pathogens, causes serious economic losses every year. However, the underlying molecular mechanism of silkworm in resistance to BmNPV infection remains unknown, even though many differentially resistant silkworm strains have been reported [2,25,26]. This suggests that there exists some kind of immune system in the silkworm organism, but this still needs further investigation. The antiviral mechanism of silkworms in response to the BmNPV infection is a complex process, and many candidate genes and proteins related to viral infection have been identified by high throughput techniques [3,6]. Clarification of these candidates’ functions will be useful to illustrate the mechanism. *Bmapaf-1* was obtained from our previous transcriptome data of the midgut of two differentially resistant silkworm strains after feeding with BmNPV [3]. However, the role of *Bmapaf-1* in the response against the BmNPV infection is still unknown.

Apoptosis is an effective immune pathway, leading to the self-destruction of the cell, playing a vital role in maintaining organism homeostasis. Furthermore, it also retains a highly evolutionarily conserved process in different species [27]. Diverse apoptotic stimuli, including viral infections, can trigger apoptosis through several apoptotic pathways, such as the mitochondrial apoptosis pathway [28]. The response mechanism of apoptosis might fit with other anti-viral mechanisms among different species based on the translational arrest [29]. Mitochondrial apoptosis has been reported to be involved in host defenses against environmental pressure by releasing Cytc into the cytoplasm to activate apoptosis, which also has been proven in silkworms in our previous report [25]. Moreover, BmCytc, released into the cytoplasm, could regulate its downstream genes to defend the BmNPV infection, including *Bmapaf-1*, *BmNc*, and *Bmcas-1*. To further confirm the apoptosis pathway, the function of *Bmapaf-1* in response against BmNPV infection was determined in this study.

Bioinformatics analysis is a useful tool for predicting gene functional information; a relatively high conservation domain of BmApaf-1 in CARD and NB-ARC with selected other species indicated that it might play an important role in the silkworm apoptosis pathway (Appendix A). This was also found in the phylogenetic tree, with BmApaf-1 and 15 other homologs all belonging to Lepidoptera (Appendix A). Besides, a unique functional domain, transposase_1, in BmApaf-1 indicated that BmApaf-1 might have a special function, which still needs to be further analyzed (Appendix A). Furthermore, the relatively high expression level of *Bmapaf-1* in the period of embryonic fast development indicated that it might be involved in embryonic development (Figure 1A). The significant high expression in pupa and adult revealed that it might be affected by ecdysone and was involved in the metamorphosis process (Figure 1B). Additionally, the significant high expression in the testis and ovary suggest potential roles in reproduction (Figure 1C).

In our previous transcriptome data, *Bmapaf-1* was identified as a candidate antiviral gene for its differential expression after BmNPV infection, which was further validated by its significantly different expression in different tissues of two different resistant strains, YeA and YeB, after inoculation with BV-EGFP at 48 h (Figure 2). These results indicated that *Bmapaf-1* had a close relationship with the virus infection. To analyze the role of *Bmapaf-1* in the mitochondrial apoptosis pathway, its two downstream homologous genes in *Drosophila* were determined after knockdown and overexpression of *Bmapaf-1* in BmN cells, that is, *BmNc* and *Bmcas-1*. The significant upregulation of *BmNc* and *Bmcas-1* in the transgenic cell line showed that *Bmapaf-1* could regulate the expression of the two genes (Figure 5), which was also validated after knockdown of *Bmapaf-1* in BmN cells (Figure 3). Therefore, *Bmapaf-1* is one of the upstream genes of *BmNc* and *Bmcas-1* in the mitochondrial apoptosis pathway.

To further study the role of *Bmapaf-1* in response to BmNPV infection, the variation of BmNPV was analyzed after knockdown and overexpression of *Bmapaf-1* in BmN cells. Green fluorescence signals of BV-EGFP were significantly stronger at 72 h after knockdown of *Bmapaf-1*, compared to the control group (Figure 4A–C), but the effect was not observed at 24 h and 48 h, indicating knockdown of *Bmapaf-1* could be beneficial for the virus infection. This phenomenon was also validated in the transgenic cell line (Figure 5A–C). Further, this phenomenon was also confirmed by the significantly higher expression level of *vp39* in the RNAi group and the significantly lower expression level in transgenic group as compared with the control group (Figure 4D and Figure 6D). To further confirm whether this phenomenon was related to apoptosis, the apoptosis inducer NSC348884 and inhibitor Z-DEVD-FMK were used. The obvious downregulation of *vp39* in the RNAi group and its negative control after treatment with the inducer NSC348884 at 72 h (Figure 7A) and the significant upregulation in the transgenic cell line and its negative control after treatment with the inhibitor Z-DEVD-FMK (Figure 7B) showed that *Bmapaf-1* was involved in the antiviral infection by regulating the mitochondrial apoptosis pathway.

The significantly different resistance in silkworms in response against BmNPV infection indicates that there indeed exists some kind of immune response. Apoptosis, as one kind of innate immune system, plays a vital role in the response against the pathogen’s infection. Based on the results in this study and a previous one [25], as well as relevant reports, we speculated that mitochondria received some kind of signal as soon as budded viruses (BVs) entered into the host cell via clathrin-mediated endocytosis [30]. These signals can change the mitochondrial membrane potential, further changing permeability of the cell membrane [31], which is beneficial for the release of BmCytc into the cytoplasm [32]. Once BmCytc is released into the cytoplasm, it will combine with BmApaf-1 [25] to be an apoptotic complex that can promote the expression of downstream BmNc. After the activation of BmNc, its downstream protein BmCas-1 will be activated, and then regulate the process of apoptosis that can be used to respond to virus replication in BmN cells (Figure 8).

## 5. Conclusions

The significantly differential expression of *Bmapaf-1* in different tissues of differential resistant strains following BmNPV infection verified the relationship between *Bmapaf-1* and BmNPV. The anti-BmNPV function of *Bmapaf-1* was confirmed by RNAi using the corresponding siRNA and overexpression using the pIZT/V5-His-mCherry insect vector harboring the *Bmapaf-1* in vitro. Further results after treatment with apoptosis inducer, NSC348884, and inhibitor, Z-DEVD-FMK, showed *Bmapaf-1* was involved in the response against BmNPV infection by the mitochondrial apoptosis pathway.

## Figures and Tables

**Figure 1 insects-11-00647-f001:**
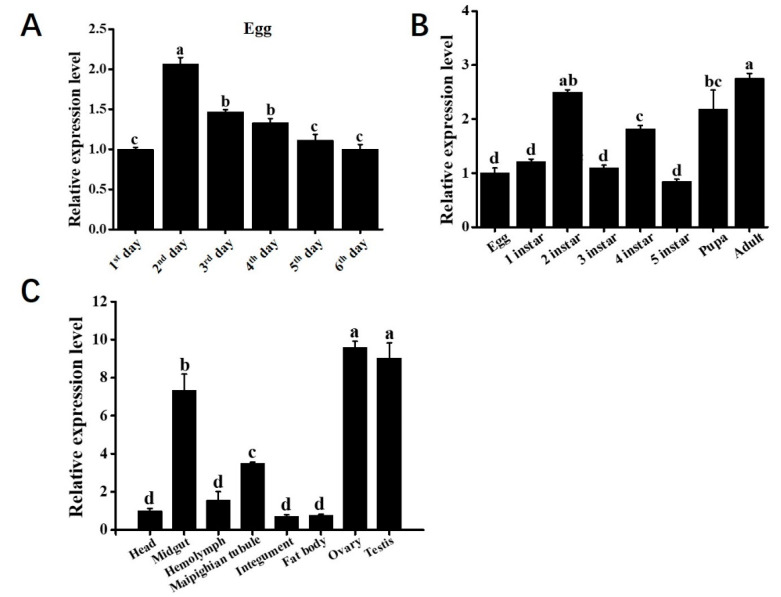
The spatiotemporal expression analysis of *Bmapaf-1* using RT-qPCR. Relative expression levels of *Bmapaf-1* among different egg development times (**A**); different developmental stages (**B**); and different tissues (**C**). 1st day, period of the longest embryo; 2nd day, period of protuberance occurred; 3rd day, prophase of protuberance rapid development; 4th day, period of shortening; 5th day, period of embryonic reversal; 6th day, head pigmentation period. *BmGAPDH* was used to normalize the data that were shown as the mean ± standard error, and the mean is from three independent repeats. The 2^−△△Ct^ method was adopted to calculate the relative expression level. Differences among triple repeats were analyzed using the SPSS Statistics 20 software (IBM, USA) with the one-way ANOVA method. Different letters represented the significant difference (a, b, c; *p* < 0.05).

**Figure 2 insects-11-00647-f002:**
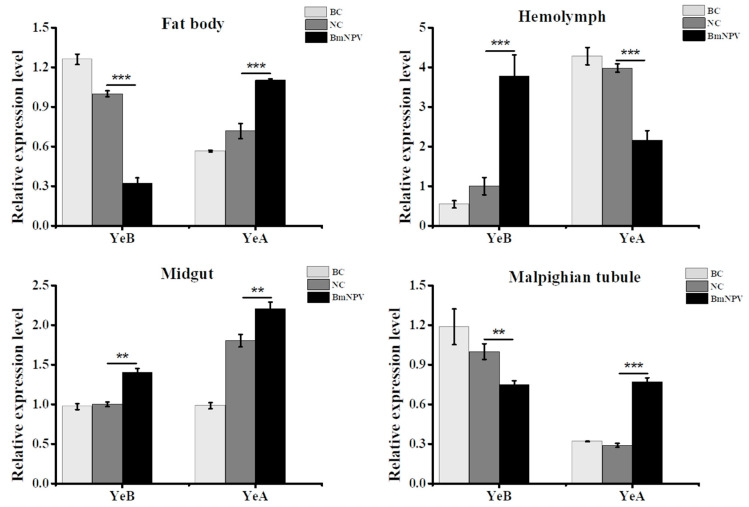
The analysis of *Bmapaf-1* expression level in different tissues of different resistant strains following BmNPV infection using RT-qPCR. YeA was a resistant silkworm strain, and YeB was a susceptible silkworm strain. BC, blank control; NC, negative control. *BmGAPDH* was used to normalize the data that were shown as the mean ± standard error, and the mean was from three independent repeats. The 2^−△△Ct^ method was adopted to calculate the relative expression level. Differences among triple repeats were analyzed using the SPSS Statistics 20 software (IBM, USA) with the one-way ANOVA method. Asterisks represented the significant difference, as follows: ** *p* < 0.01; *** *p* < 0.001.

**Figure 3 insects-11-00647-f003:**
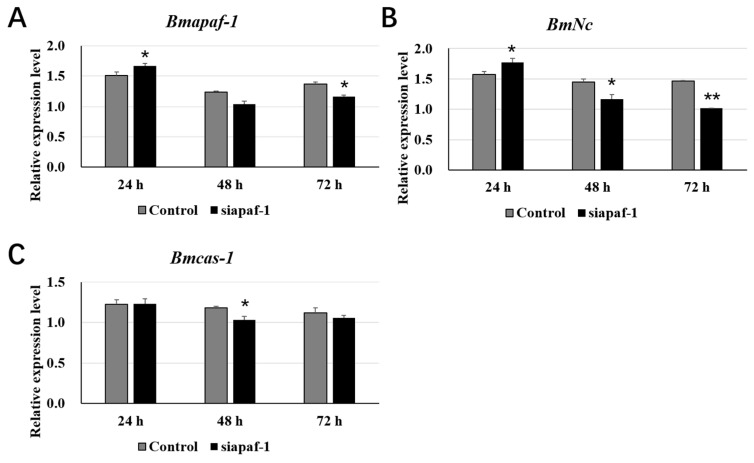
Expression analysis of selected downstream genes after knockdown of *Bmapaf-1* with siRNA at different time points using RT-qPCR. (**A**) Expression level analysis of *Bmapaf-1* after transfection with siRNA at 24, 48, and 72 h. Analysis of *BmNc* (**B**) and *Bmcas-1* (**C**) expression levels after knockdown of *Bmapaf-1* at different time points. *BmGAPDH* was used to normalize the data that were showed as the mean ± standard error, and the mean was from three independent repeats. The 2^−△△Ct^ method was adopted to calculate the relative expression level. Differences among triple repeats were analyzed using the SPSS Statistics 20 software (IBM, USA) with the one-way ANOVA method. Asterisks represent the significant difference, as follows: * *p* < 0.05; ** *p* < 0.01.

**Figure 4 insects-11-00647-f004:**
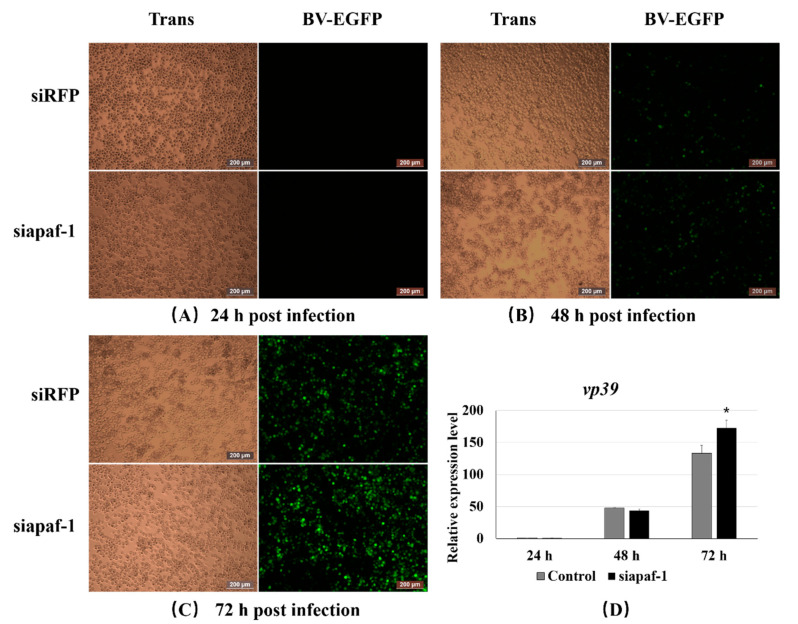
Analysis of BmNPV infection following knockdown of *Bmapaf-1* at different times in BmN cells: 24 h (**A**); 48 h (**B**), and 72 h (**C**) after BV-EGFP infection. (**D**) The analysis of *vp39* expression after knockdown of *Bmapaf-1* at different times. Scale bar = 200 μm. Trans (white), optical transmission. EGFP (green), expressed following the replication of BV. *BmGAPDH* was used to normalize the data that were showed as the mean ± standard error, the mean was from three independent repeats. The 2^−△△Ct^ method was adopted to calculate the relative expression level. Differences among triple repeats were analyzed using the SPSS Statistics 20 software (IBM, USA) with the one-way ANOVA method. Asterisks represent the significant difference, * *p* < 0.05.

**Figure 5 insects-11-00647-f005:**
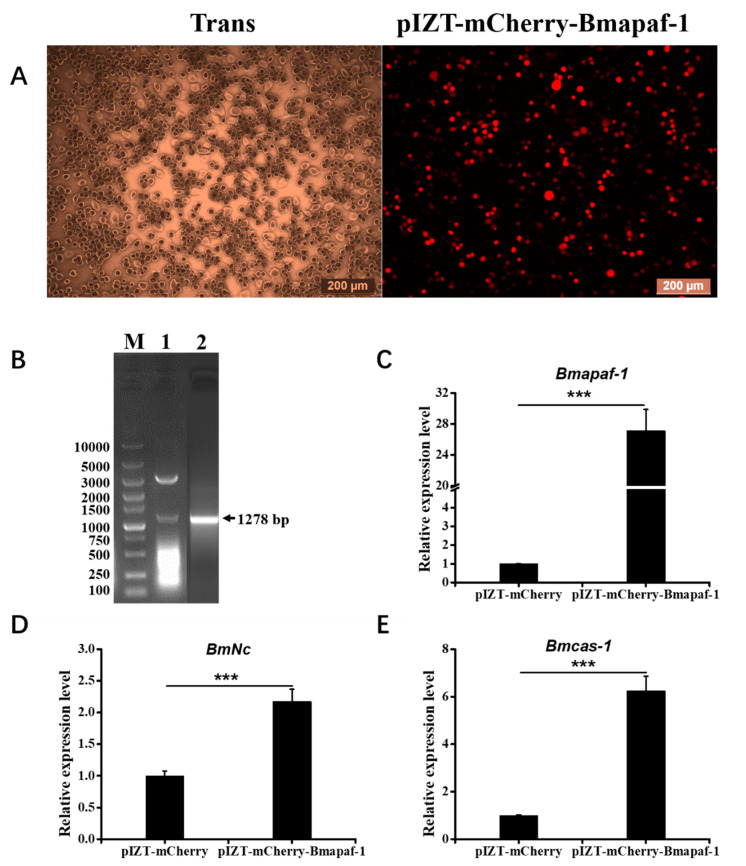
Expression analysis of selected downstream genes after overexpression of *Bmapaf-1* at different times in BmN cells. (**A**) Overexpression detection of *Bmapaf-1* after transfection with pIZT-mCherry-Bmapaf-1 in BmN cells. Scale bar = 200 μm. Trans (white), optical transmission. mCherry (Red), fused expression with Bmapaf-1 protein. (**B**) The construction of pIZT-mCherry-Bmapaf-1: (1) validation of the recombinant vector using double enzyme digestion and (2) amplification of the functional domain of *Bmapaf-1*. (**C**) The expression level analysis of *Bmapaf-1* after transfecting with recombinant vector using RT-qPCR. Expression level analysis of *BmNc* (**D**) and *Bmcas-1* (**E**) after overexpression of *Bmapaf-1*. *BmGAPDH* was used to normalize the data that were shown as the mean ± standard error, and the mean was from three independent repeats. The 2^−△△Ct^ method was adopted to calculate the relative expression level. Differences among triple repeats were analyzed using the SPSS Statistics 20 software (IBM, USA) with the one-way ANOVA method. Asterisks represent the significant difference, as follows: *** *p* < 0.001.

**Figure 6 insects-11-00647-f006:**
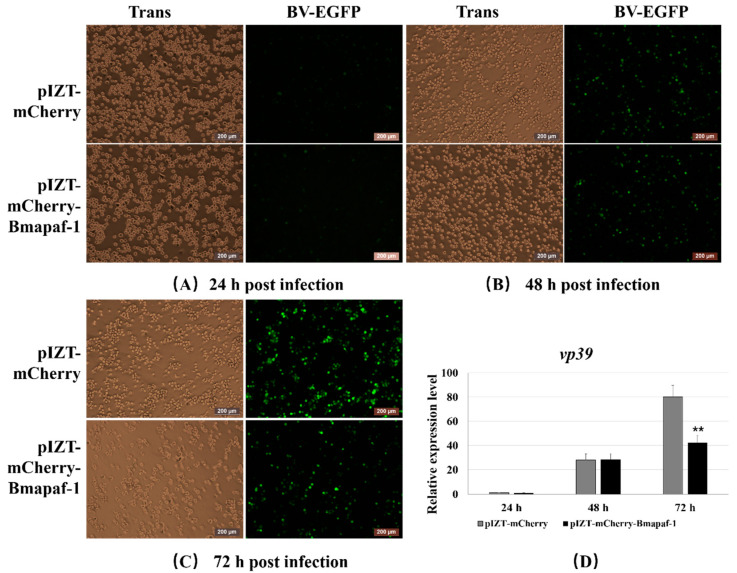
Analysis of BmNPV infection after overexpression of *Bmapaf-1* at different times in BmN cells, 24 h (**A**), 48 h (**B**), and 72 h (**C**), after BV-EGFP infection. (**D**) The analysis of *vp39* expression after overexpression of *Bmapaf-1* at different times. Scale bar = 200 μm. Trans (white), optical transmission. EGFP (green), expressed following the replication of BV. *BmGAPDH* was used to normalize the data that were shown as the mean ± standard error, the mean was from three independent repeats. The 2^−△△Ct^ method was adopted to calculate the relative expression level. Differences among triple repeats were analyzed using the SPSS Statistics 20 software (IBM, USA) with the one-way ANOVA method. Asterisks represent the significant difference, as follows: ** *p* < 0.01.

**Figure 7 insects-11-00647-f007:**
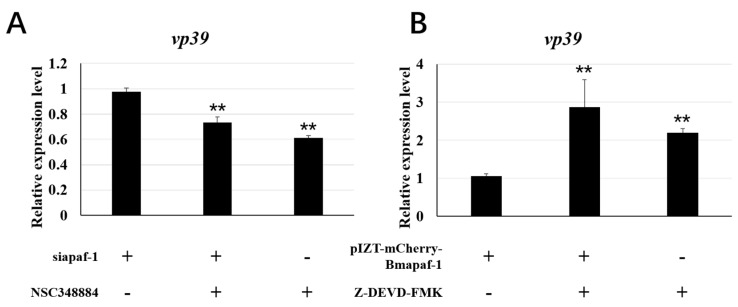
The analysis of BmNPV replication in the RNAi and overexpression group following regulation of apoptosis in BmN cells using RT-qPCR. (**A**) The replication of BmNPV in the RNAi group after treatment with inducer NSC348884. (**B**) The analysis of BmNPV replication in the transgenic BmN cell line following treatment with apoptosis inhibitor Z-DEVD-FMK. *BmGAPDH* was used to normalize the data that were shown as the mean ± standard error, the mean was from three independent repeats. The 2^−△△Ct^ method was used to calculate the relative expression level. Differences among triple repeats were analyzed using the SPSS Statistics 20 software (IBM, USA) with the one-way ANOVA method. Asterisks represented the significant difference, as follows: ** *p* < 0.01.

**Figure 8 insects-11-00647-f008:**
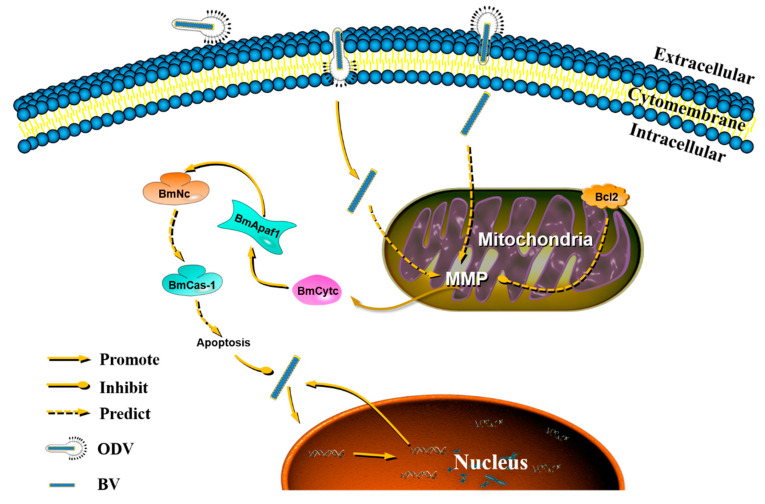
The proposed function of BmApaf-1 in response against BmNPV infection by the mitochondrial apoptosis pathway. MMP, mitochondrial membrane potential.

**Table 1 insects-11-00647-t001:** List of primers used in this study.

Genes Name	Forward Primers (5′-3′)	Revers Primers (5′-3′)
*Bmapaf-1*	TCACAACCCTCTAAAATCACACCAG	CGACAGCCAGTAATGGGTGTATGAG
*BmNc*	GAGGACGATGTGAGCAGGGAT	TTCAGCAGGAACGAAATGTAGC
*Bmcas-1*	AACGGCAATGAAGACGAAGG	GGTGCCCGTGCGAGATTTTA
*BmGAPDH*	CCGCGTCCCTGTTGCTAAT	CTGCCTCCTTGACCTTTTGC
*VP39*	CAACTTTTTGCGAAACGACTT	GGCTACACCTCCACTTGCTT
*Bmapaf-1 KE*	GGGGTACCAGGAAGCTGCTGCAGCA	CGGAATTCTATGTTTTCGACTTCGTTGAC

**Table 2 insects-11-00647-t002:** Primers used to synthesize siRNA.

Primer Names	Sequences (5′-3′)
Bmapaf-1-1 Olig-1	GATCACTAATACGACTCACTATAGGGGCTAATCTGGTCATAGTTATT
Bmapaf-1-1 Olig-2	AATAACTATGACCAGATTAGCCCCTATAGTGAGTCGTATTAGTGATC
Bmapaf-1-1 Olig-3	AAGCTAATCTGGTCATAGTTACCCTATAGTGAGTCGTATTAGTGATC
Bmapaf-1-1 Olig-4	GATCACTAATACGACTCACTATAGGGTAACTATGACCAGATTAGCTT
Bmapaf-1-2 Olig-1	GATCACTAATACGACTCACTATAGGGGCTAATTATCACCCGCAAATT
Bmapaf-1-2 Olig-2	AATTTGCGGGTGATAATTAGCCCCTATAGTGAGTCGTATTAGTGATC
Bmapaf-1-2 Olig-3	AAGCTAATTATCACCCGCAAACCCTATAGTGAGTCGTATTAGTGATC
Bmapaf-1-2 Olig-4	GATCACTAATACGACTCACTATAGGGTTTGCGGGTGATAATTAGCTT
RFP-Olig-1	GATCACTAATACGACTCACTATAGGGGCACCCAGACCATGAGAATTT
RFP-Olig-2	AAATTCTCATGGTCTGGGTGCCCCTATAGTGAGTCGTATTAGTGATC
RFP-Olig-3	AAGCACCCAGACCATGAGAATCCCTATAGTGAGTCGTATTAGTGATC
RFP-Olig-4	GATCACTAATACGACTCACTATAGGGATTCTCATGGTCTGGGTGCTT

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
