# Peer review of "Bmapaf-1 is Involved in the Response against BmNPV Infection by the Mitochondrial Apoptosis Pathway"

_insects, 2020, doi:10.3390/insects11090647_

Round 1
Reviewer 1 Report
Based on expression profiling and functional characterization of the Bombyx apoptosis protease-activating factor 1, Wang et al conclude that the gene functions in the mitochondrial apoptotic pathway and contributes to BmNPV resistance in Bombyx. Although the methods used are appropriate and the data shown are largely consistent with the author conclusions, more methodological details and/or clarifications should be included as some expected controls are missing. Specific comments that the authors might consider are listed below.
Major comments
1) The manuscript would greatly benefit from more thorough English editing. As currently written, the somewhat confused word usage and sentence structure can impede reader comprehension.
Examples –
Line 182 – “…contained a unique functional domain…”. Do the authors mean that the transposase-1 domain is unique or that the domain is unique to Bombyx Apaf-1?
Line 276 – “The functional domain of Bmapaf-1 was inserted…”. Do the authors mean that a single functional domain was inserted into the expression vector or that the Bombyx Apaf-1 ORF was cloned in?
Note- these are just two examples of the awkward word usage and sentence structure found throughout the manuscript.
2) Provide more methodological details.
- Line 92 – The statement that the EGFP sequence was inserted into the BmNPV genome between the “BamHI and XhoI” sites is too vague. Provide more details on the insertion point. It is unclear if the authors are generating a chimeric protein tagged with the EGFP marker or if the EGFP sequence has been inserted under the control of the polyhedron promoter.
- Line 97 – Indicate the accession number of the Bombyx protein sequence used in the bioinformatic analyses.
- Line 101 – Provide a supplementary table listing the accession numbers of the proteins used in the phylogenetic analyses. Also, inclusion of an outgroup, perhaps the Drosophila Apaf, would generate a more robust tree.
- Line 106 – It is unclear which strain of larvae was used. Were all three strains listed on line 84 used?
- Line 110- It is unclear in this section, as well as throughout the manuscript, how many biological replicates were used. Furthermore, when replicates are mentioned it is unclear if the authors are referring to technical replicates or biological replicates. The preference would be for three technical replicates (eg. qPCR) for three biological replicates.
- Line 116 – What amount of RNA was used in the cDNA synthesis? Also, the PrimeScript RT kit comes with two primers, random hexamers and oligo-dT. Which was used in the synthesis?
- Line 128 – It is unclear if amplification efficiencies were determined for the primers used. More details should be provided in terms of how efficiencies were determined and what the respective values were for each primer pair.
- Line 162 – How long was the transfection period?
- Line 163 – At what point in time post-transfection were the transfected cells assayed?
- Line 167 – Neither of the compounds listed are water soluble. Indicate what reagent (DMSO?) and its percentage was used as a vehicle to deliver the compounds to the cells. In addition, a vehicle alone control should be included in the data analyses.
3) Provide more clarity on the purpose of using the P50 strain for transcript expression profiling when the Introduction indicated that the target gene was identified via RNA-Seq analysis of resistant strains. Why not profile transcript expression in the YeB (susceptible) and YeA (resistant) strains as done in Figure 2? The use of the P50 strain seems incongruous with the other results.
4) It is unclear what “control” was used in Figure 2. Furthermore, given that the particular study was looking at the transcriptional effects of BmNPV “inoculation” on Bmapaf-1 expression one would expect to see results for untreated larvae, mock-treated larvae, and then BmNPV-treated larvae. However, the data shown has only one control.
5) In Figure 3, why is there a significant upregulation in the target gene and one of the downstream components (BmNc) at 24 h post-siRNA treatment? Has this pathway been implicated in an anti-RNAi response? While I appreciate the difficulty in determining transcript levels of genes in a pathway, especially those that may be involved in multiple non-overlapping responses, the explanation provided for the varied responses in transcript levels over time for Bmapaf-1, BmNc, and Bmcas-1 could be clarified better. In addition, line 243 states that Bmcas-1 expression was upregulated at 48 h post-siRNA treatment; however, the data shown indicates significant reduction in transcript levels.
6) In Figure 7 A, additional data for siapf negative cells with NSC34884 should be shown. Similarly, data for BmN cells alone should be included in Fig 7B.
7) The Discussion could be expanded to provide more details on the potential signals that trigger the apoptotic response and how conserved the pathway may be in other lepidopterans (ie is the mechanism used to drive NPV resistance in other lepidopterans or is it perhaps specific to Bombyx?). How would the proposed mechanism fit with other NPV resistance mechanisms such as translational arrest (see Nagamine & Sako, 2016, PLOS ONE)?
8) The authors state in line 365 that the transposase-1 domain that appears to be unique to the Bombyx protein may have some unique functionality. It seems that that it would be rather simple to address this question (or at least begin to provide insights) by performing the mCherry tagged experiments using a truncated version of Bmapaf-1 in which the transposase-1 domain is deleted.
Minor comments –
Lines 47-48 – Perhaps provide more than one reference for each of the methods used to examine the silkworm antiviral response
Line 135 – For better clarity perhaps indicate that the siRNA oligonucleotides are templates for subsequent T7 based transcription.
Line 177 – Define the domain abbreviations and indicate their typical functional roles and why those domains are important for the gene in question. As written, unless the readers have a background in apoptosis, the term “CARD” likely has little meaning.
Line 349 – A reference for the transcriptomic study examining the effects of BmNPV on gene expression in the midgut of resistant strains should be provided.
Author Response
Thank you for your professional suggestion, we have revised our manuscript after having carefully considered the comments, details are as follows:
Major comments:
- The manuscript would greatly benefit from more thorough English editing. As currently written, the somewhat confused word usage and sentence structure can impede reader comprehension.
Reply: We have asked the PRS institute to revise our manuscript again, and we also have carefully checked all the content of our manuscript and revised the improper description by ourselves.
- Line 92 – The statement that the EGFP sequence was inserted into the BmNPV genome between the “BamHI and XhoI” sites is too vague. Provide more details on the insertion point. It is unclear if the authors are generating a chimeric protein tagged with the EGFP marker or if the EGFP sequence has been inserted under the control of the polyhedron promoter.
Reply: The EGFP gene was insterted in the BamH I and Xho I sites of the plasmid pFASTbac1 to generate a recombinant virus to express the EGFP protein under the polyhedron promoter. The description has been revised (Line 93).
- Line 97 – Indicate the accession number of the Bombyx protein sequence used in the bioinformatic analyses.
Reply: The accession number has been added (Line 102).
- Line 101 – Provide a supplementary table listing the accession numbers of the proteins used in the phylogenetic analyses. Also, inclusion of an outgroup, perhaps the Drosophila Apaf, would generate a more robust tree.
Reply: The accession numbers of the proteins used in the phylogenetic analyses have been added into the supplementary material Table S1. We chose the Bicyclus anynana as an outgroup in Fig. S2 in the supplementary material that we sent for review. Besides, we have tried to add Drosophila Apaf, but the low identity caused an error in building the tree.
- Line 106 – It is unclear which strain of larvae was used. Were all three strains listed on line 84 used?
Reply: No, just YeA and YeB, which has been revised (Line 110).
- Line 110- It is unclear in this section, as well as throughout the manuscript, how many biological replicates were used. Furthermore, when replicates are mentioned it is unclear if the authors are referring to technical replicates or biological replicates. The preference would be for three technical replicates (eg. qPCR) for three biological replicates.
Reply: In this study, three biological replicates were used to eliminate error, which has been added into the manuscript (Line 115).
- Line 116 – What amount of RNA was used in the cDNA synthesis? Also, the PrimeScript RT kit comes with two primers, random hexamers and oligo-dT. Which was used in the synthesis?
Reply: The first strand cDNA was synthesized with 1.0 μg RNA. The PrimeScript RT kit contains Oligo dT Primer and random 6 mers. The description has been revised (Line 122).
- Line 128 – It is unclear if amplification efficiencies were determined for the primers used. More details should be provided in terms of how efficiencies were determined and what the respective values were for each primer pair.
Reply: The amplification efficiency of all primers was tested, and more than 95% efficiency was used for further study. The unclear description has been revised (Line 127).
- Line 162 – How long was the transfection period?
Reply: According to the manufacturer’s instructions, the best effective time for the transfection reagent is 24 h after transfection. It has been added into line 171.
- Line 163 – At what point in time post-transfection were the transfected cells assayed?
Reply: To analyzed the change of virus infection, the fluorescence signal of virus were analyzed at 24, 48, 72 h post-inoculation after knockdown and overexpression of Bmapaf-1 24 h. The description has been revised (Line 173).
- Line 167 – Neither of the compounds listed are water soluble. Indicate what reagent (DMSO?) and its percentage was used as a vehicle to deliver the compounds to the cells. In addition, a vehicle alone control should be included in the data analyses.
Reply: Both of the compound were resolved in DMSO to get 1.0 mM mother solution, which has been added in the line of 179.
- Provide more clarity on the purpose of using the P50 strain for transcript expression profiling when the Introduction indicated that the target gene was identified via RNA-Seq analysis of resistant strains. Why not profile transcript expression in the YeB (susceptible) and YeA (resistant) strains as done in Figure 2? The use of the P50 strain seems incongruous with the other results.
Reply: The p50 silkworm is a strain widely used in different laboratories, and its genome has been sequenced. The analysis of Bmapaf-1 in p50 can provide a relative better data reference for readers and database supplement. However, YeB (susceptible) and YeA (resistant) strains are two very special strains in our laboratory, the profile of Bmapaf-1 can not be repeated in other laboratories very well. The purpose of using the p50 strain has been added in line 210.
The use of the p50 strain seems incongruous with the other results, which might be caused by the different genetic background among different silkworm strains. Besides, the purpose of analysis the expression pattern of Bmapaf-1 in YeA and YeB following BmNPV inoculation is to clear the relationship between Bmapaf-1 and BmNPV infection, and the results were very clear.
- It is unclear what “control” was used in Figure 2. Furthermore, given that the particular study was looking at the transcriptional effects of BmNPV “inoculation” on Bmapaf-1 expression one would expect to see results for untreated larvae, mock-treated larvae, and then BmNPV-treated larvae. However, the data shown has only one control.
Reply: The control is the mock-treated larvae, and the description has been added in line 230. Moreover, the black control has been added in Fig. 2.
- In Figure 3, why is there a significant upregulation in the target gene and one of the downstream components (BmNc) at 24 h post-siRNA treatment? Has this pathway been implicated in an anti-RNAi response? While I appreciate the difficulty in determining transcript levels of genes in a pathway, especially those that may be involved in multiple non-overlapping responses, the explanation provided for the varied responses in transcript levels over time for Bmapaf-1, BmNc, and Bmcas-1 could be clarified better. In addition, line 243 states that Bmcas-1 expression was upregulated at 48 h post-siRNA treatment; however, the data shown indicates significant reduction in transcript levels.
Reply: We also confuse about the result when we got the data, the reason might be the one you said that multiple non-overlapping responses. But on the other hand, the consistent upregulation and downregulation at the same time point hints the regulation of Bmapaf-1 to its downstream genes. Besides, we are sorry for the wrong description, which has been revised.
- In Figure 7 A, additional data for siapf negative cells with NSC34884 should be shown. Similarly, data for BmN cells alone should be included in Fig 7B.
Reply: The negative control has been added in Figure 7.
- The Discussion could be expanded to provide more details on the potential signals that trigger the apoptotic response and how conserved the pathway may be in other lepidopterans (ie is the mechanism used to drive NPV resistance in other lepidopterans or is it perhaps specific to Bombyx?). How would the proposed mechanism fit with other NPV resistance mechanisms such as translational arrest (see Nagamine & Sako, 2016, PLOS ONE)?
Reply: The discussion has been added based on your recommend (Line 380).
- The authors state in line 365 that the transposase-1 domain that appears to be unique to the Bombyx protein may have some unique functionality. It seems that it would be rather simple to address this question (or at least begin to provide insights) by performing the mCherry tagged experiments using a truncated version of Bmapaf-1 in which the transposase-1 domain is deleted.
Reply: Thank you for your professional suggestion, we will try to do it and present the result in our next study that uses CRISPR/Cas9 system to further verify the function of Bmapaf-1 in vivo.
Minor comments:
- Lines 47-48 – Perhaps provide more than one reference for each of the methods used to examine the silkworm antiviral response
Reply: References have been added.
- Line 135 – For better clarity perhaps indicate that the siRNA oligonucleotides are templates for subsequent T7 based transcription.
Reply: The description was added in line 143 and 145.
- Line 177 – Define the domain abbreviations and indicate their typical functional roles and why those domains are important for the gene in question. As written, unless the readers have a background in apoptosis, the term “CARD” likely has little meaning.
Reply: The define of “CARD” abbreviation has been added.
- Line 349 – A reference for the transcriptomic study examining the effects of BmNPV on gene expression in the midgut of resistant strains should be provided.
Reply: The reference has been added.
Reviewer 2 Report
The authors examined whether Bmapaf-1 is involved in BmNPV infection by regulating the mitochondrial apoptosis pathway. First, the expression of Bmapaf-1 was knocked down and overexpressed using siRNA and an expression vector, respectively, and the variation of BmNPV was recorded and determined with fluorescence microscopy with RT-qPCR. Finally, the relationship between Bmapaf-1 and apoptosis was analyzed using an apoptosis inducer and an inhibitor. The authors showed circumstantial evidence that suggests that Bmapaf-1 is involved in BmNPV infection by regulating apoptosis.
Major points:
Description of the interpretation of the results of the spatiotemporal expression pattern of Bmapaf-1 in the Results and Discussion sections are abstractive and too short. Therefore, it is hard to obtain valid information or concrete conclusion from the results. The authors should try to describe in detail the implication of the differences of Bmapaf-1 expression level among tissues and among B. mori strains.
Line 321-339: I feel that the authors should have included a control in the experiments shown in Figs. 7 A and B, i.e., siapaf-1 (-) with NSC348884 (+) and pIZT-mCherry-Bmapaf-1(-) with Z-DEVD-FMK (+), respectively. This would reinforce the conclusion that Bmapaf-1 is involved in apoptosis.
Minor points:
Line 64-67: It would be better to show the reference.
Line 90: “EGFP” here should be defining by full spelling because the abbreviation firstly appears here. In relation to this, the full spelling in line 255 should be deleted.
Line 116-117: The English of this sentence should be checked.
Line 154-156: This sentence seems awkward.
Line 329: Fig.7A should be Fig. 7B.
Author Response
Thank you for your professional suggestion, we have revised our manuscript after having carefully considered the comments, details are as follows:
Major concerns:
- Description of the interpretation of the results of the spatiotemporal expression pattern of Bmapaf-1 in the Results and Discussion sections are abstractive and too short. Therefore, it is hard to obtain valid information or concrete conclusion from the results. The authors should try to describe in detail the implication of the differences of Bmapaf-1 expression level among tissues and among B. mori strains.
Reply: The description of the spatiotemporal expression pattern of Bmapaf-1 was revised in detail (Line 393-397).
- Line 321-339: I feel that the authors should have included a control in the experiments shown in Figs. 7 A and B, i.e., siapaf-1 (-) with NSC348884 (+) and pIZT-mCherry-Bmapaf-1(-) with Z-DEVD-FMK (+), respectively. This would reinforce the conclusion that Bmapaf-1 is involved in apoptosis.
Reply: The control has been done and added into Fig. 7.
Minor comments:
- Line 64-67: It would be better to show the reference.
Reply: The reference has been added.
- Line 90: “EGFP” here should be defining by full spelling because the abbreviation firstly appears here. In relation to this, the full spelling in line 255 should be deleted.
Reply: It has been revised.
- Line 116-117: The English of this sentence should be checked.
Reply: It has been revised.
- Line 154-156: This sentence seems awkward.
Reply: It has been revised.
- Line 329: Fig.7A should be Fig. 7B.
Reply: It has been revised.
Round 2
Reviewer 1 Report
The authors' consideration for the reviewer comments/suggestions are appreciated. Most of my comments, in particular those concerning the absence of controls in the Figures, have been sufficiently addressed. Some language issues, however, remain. Please see my suggested edits below.
Line 123 – was repeated using three biological replicates
Line 151 – as the siapaf-1 template for siRNA synthesis using the In Vitro Transcription T7 kit (TaKaRa, Japan)…
Line 166 – BmN cells, which are derived from the silkworm ovary, were cultured in TC-100 media (AppliChem, Germany)…
Line 174 – The best transfection efficiency was obtained at 24 h, as such that time point was selected for all subsequent analyses.
Line 182 – Both of the compounds were dissolved in DMSO to generate a 1 mM working solution.
Line 233 - …and the blank control is no injection.
Line 246 – blank control
Line 384 – Mitochondrial apoptosis has been reported…
Line 399 - …might be affected by ecdysone…
Line 400 - …ovary suggest potential roles in reproduction
Author Response
Thank you for your professional suggestion, we have revised our manuscript based on your suggestions, details are as follows:
Minor comments:
- Line 123 – was repeated using three biological replicates
Reply: It was revised.
- Line 151 – as the siapaf-1 template for siRNA synthesis using the In Vitro Transcription T7 kit (TaKaRa, Japan)…
Reply: It was revised.
- Line 166 – BmN cells, which are derived from the silkworm ovary, were cultured in TC-100 media (AppliChem, Germany)…
Reply: It was revised.
- Line 174 – The best transfection efficiency was obtained at 24 h, as such that time point was selected for all subsequent analyses.
Reply: It was revised.
- Line 182 – Both of the compounds were dissolved in DMSO to generate a 1 mM working solution.
Reply: It was revised.
- Line 233 - …and the blank control is no injection.
Reply: It was revised.
- Line 246 – blank control
Reply: It was revised.
- Line 384 – Mitochondrial apoptosis has been reported…
Reply: It was revised.
- Line 399 - …might be affected by ecdysone…
Reply: It was revised.
- Line 400 - …ovary suggest potential roles in reproduction
Reply: It was revised.
Reviewer 2 Report
The authors addressed my suggestions. I only suggest minor revision below.
Line 130–131: The sentence “The synthetic ---" seems awkward. “The quality of the synthetic ---” may be better?
Line 167–168: I feel the phrase “the mixture medium that contained --- streptomycin)” is awkward. Would it be better that” the medium TC-100 (AppliChem, Germany) that contained 10% (v/v) fetal bovine serum (FBS; Thermo Scientific, USA) and 1% double-155 antibiotics (penicillin and streptomycin)”?
Lines 250, 325, 346 and 366: “*p <0.05;” should be deleted because the one asterisk does not appear in the figures.
Author Response
Thank you for your professional suggestion, we have revised our manuscript based on your suggestions, details are as follows:
Minor comments:
- Line 130–131: The sentence “The synthetic ---" seems awkward. “The quality of the synthetic ---” may be better?
Reply: It was revised.
- Line 167–168: I feel the phrase “the mixture medium that contained --- streptomycin)” is awkward. Would it be better that” the medium TC-100 (AppliChem, Germany) that contained 10% (v/v) fetal bovine serum (FBS; Thermo Scientific, USA) and 1% double-155 antibiotics (penicillin and streptomycin)”?
Reply: It was revised.
- Lines 250, 325, 346 and 366: “*p <0.05;” should be deleted because the one asterisk does not appear in the figures.
Reply: These were revised.